# Point-of-care bulk testing for SARS-CoV-2 by combining hybridization capture with improved colorimetric LAMP

Lukas Bokelmann[1], Olaf Nickel[2], Tomislav Maricic [1], Svante Pääbo [1,3], Matthias Meyer[1], Stephan Borte[2,4,5] & Stephan Riesenberg [1✉]

Efforts to contain the spread of SARS-CoV-2 have spurred the need for reliable, rapid, and cost-effective diagnostic methods which can be applied to large numbers of people. However, current standard protocols for the detection of viral nucleic acids while sensitive, require a high level of automation and sophisticated laboratory equipment to achieve throughputs that allow whole communities to be tested on a regular basis. Here we present Cap-iLAMP (capture and improved loop-mediated isothermal amplification) which combines a hybridization capture-based RNA extraction of gargle lavage samples with an improved colorimetric RT-LAMP assay and smartphone-based color scoring. Cap-iLAMP is compatible with point-of-care testing and enables the detection of SARS-CoV-2 positive samples in less than one hour. In contrast to direct addition of the sample to improved LAMP (iLAMP), Cap-iLAMP prevents false positives and allows single positive samples to be detected in pools of 25 negative samples, reducing the reagent cost per test to ~1 Euro per individual.

[1] Max Planck Institute for Evolutionary Anthropology, Leipzig, Germany. [2] Department of Laboratory Medicine, Hospital St. Georg, Leipzig, Germany. [3] Okinawa Institute of Science and Technology, Onna-son, Japan. [4] ImmunoDeficiency Center Leipzig (IDCL) at Hospital St. Georg Leipzig, Jeffrey Modell Diagnostic and Research Center for Primary Immunodeficiency Diseases, Leipzig, Germany. [5] Division of Clinical Immunology, Department of Laboratory Medicine, Karolinska Institutet at Karolinska University Hospital Huddinge, Stockholm, Sweden. ✉email: stephan_riesenberg@eva.mpg.de

The recent global outbreak of coronavirus disease 2019 (COVID-19) has led governments to take drastic measures to contain the spread of the severe acute respiratory syndrome coronavirus 2 (SARS-CoV-2). Lock-down measures and restrictions on travel, public gatherings, and the closing of institutions such as schools, kindergartens, and universities are usually implemented broadly, affecting the lives of infected and uninfected people alike. Reliable and economical mass testing approaches for decentralized point-of-care identification of infected individuals could greatly help to achieve more directed and efficient containment of SARS-CoV-2.

Reverse transcription followed by quantitative PCR (RT-qPCR) is the most widely used method to detect RNA viruses such as SARS-CoV-2. However, its need for expensive bulky instrumentation and shortages of resources for RNA purification has spurred the search for viable alternatives even though sensitive RNA-extraction-free SARS-CoV-2 RT-qPCR-based tests are now established[1]. Loop-mediated isothermal amplification (LAMP) can rapidly amplify target nucleic acid sequences under isothermal conditions[2], and has been applied for molecular diagnostics[3]. The reaction requires four to six primers and produces concatemers of double-stranded amplification products. These can be detected directly, using intercalating dyes (e.g., SYBR green, SYTO-dyes), by triphenylmethane dye precursors and acid hydrolysis[4,5], or by cleavage with CRISPR enzymes coupled with lateral flow color detection of the cleavage product[6]. However, these methods require an opening of the tube after the reaction, thus posing the threat of cross-contaminating future reactions with the amplified product. Amplification can also be detected indirectly by hydroxynaptholblue or phenol red-based detection of the release of protons and/or pyrophosphate generated during DNA synthesis[7]. Recently, a number of studies explored ways to detect SARS-CoV-2 RNA using RT-LAMP[8–10] but they required time-consuming RNA isolation steps before the reaction. There have also been attempts to add sample directly into the reaction without prior purification. However, the pH of nasopharyngeal swab samples often varies and can adversely affect readouts[11,12].

In this work, we describe a method to detect SARS-CoV-2 RNA of a single-infected individual within a bulk sample composed of up to 26 individual patient samples by combining a hybridization capture-based RNA-extraction approach with smartphone app-assisted colorimetric detection of RT-LAMP products, a procedure that can be performed in <1 h (Fig. 1A).

We compare this method to standard extraction RT-qPCR protocols and validate its performance on 555 gargle lavage samples from a hospital, a nursing home previously affected by COVID-19, and round-robin samples from a reference institution of the German Medical Association. Compared to other point-of-care tests, we find the herein described method to be similarly specific and more sensitive.

## Results

**Development of Cap-iLAMP**. We evaluated three published RT-LAMP primer combinations targeting either the *Orf1a* gene or the *N* gene of the SARS-CoV-2 genome[9,10] using a dilution series of synthetic viral RNA (Twist Biosciences, San Francisco, CA, USA) and chose the two most sensitive primer sets (CV1-6 and CV15-20, Supplementary Table 1) to detect the *Orf1a* gene and the *N* gene, respectively, for further testing (Fig. 1B and Supplementary Fig. 1). Both primer sets could detect 500 synthetic viral RNA copies after 25–30 min incubation at 65 °C as measured by fluorescence real-time RT-LAMP (Supplementary Fig. 1A, C). Combining the primers for the *Orf1a* gene and the *N* gene in one reaction did not increase sensitivity (Supplementary Fig. 1D).

Amplification in LAMP reactions is often detected colorimetrically by a pH-sensitive dye that changes color when extensive DNA synthesis lowers the pH of the reaction[7]. As noted in a previous study[11], biological samples such as nasopharyngeal swab eluates may change the pH when added to the LAMP reaction directly, leading to false-positive results. We found that nasopharyngeal swab eluates tend to be more acidic than gargle lavage samples based on the pH-dye color change in the LAMP mix and that adding gargle lavage directly to a LAMP reaction at a final concentration of 5% leads to false-positive results in 4.7% of cases even before the isothermal incubation (Supplementary Fig. 2A, B).

It would therefore be preferable to detect the amplification product directly rather than indirectly by a pH change. To achieve this, we deposited a drop of 0.5 microliters of a 10,000× concentrated solution of the dye SYBR Green I to the cap of the tube before the reaction is initiated. A similar closed tube approach has been described for the detection of yam mosaic virus[13]. Shaking the tube after the isothermal amplification mixes the SYBR Green I with the LAMP components, stops the reaction, and allows visual detection of SARS-CoV-2 via a color change from orange/red to intense yellow (Fig. 1C). The color of the reaction can be quantified as a single numerical hue value, that is insensitive to light intensity changes and can be derived from the red, green, blue (RGB) color model[14,15], by using freely available "camera color picker" smartphone apps. We used the "Palette Cam" app (Alexander Mathers, App Store) for extracting RGB values before conversion to hue. We tested the influence of two smartphone models (Redmi 7 and iPhone 5), as well as two different light sources (daylight and fluorescent tube light) on the obtained hue value. While both smartphones result in comparable hue values, daylight resulted in a clearer separation between negative and positive samples, than fluorescent tube light (Supplementary Fig. 3A, B). We thus acquired all images using daylight. An additional advantage is that detection using SYBR Green I allows us to include acidifying enhancing enzymes in the LAMP reaction, namely Tte UvrD helicase, which prevents unspecific late amplification of artefacts resulting from primer interactions (Supplementary Fig. 4A, B) and thermostable inorganic pyrophosphatase which increases reaction speed[4].

However, the addition of patient gargle lavage directly to the improved LAMP (iLAMP) reaction resulted in false positives (13.5 %) (Supplementary Fig. 5A). Presumably, this unspecific amplification is due to DNA from the oral microbiome, food, or host cells as it can be prevented by prior λ exonuclease treatment that preferentially digests 5′-phosphorylated DNA leaving non-phosphorylated primers and iLAMP product intact (Supplementary Fig. 5B). Furthermore, gargle lavages from seven infected patient samples yielded false-negative results. We, therefore, employed a Quick extract (Lucigen, Middleton, WI, USA) lysis[16] to release more viral RNA from the capsid. But this still resulted in one false negative out of the seven gargle lavages from infected patients and the single SARS-CoV-2-negative sample was falsely positive (Supplementary Fig. 5B). We, therefore, employed a rapid (15 min) bead-capture enrichment purification akin to mRNA isolation, using two oligonucleotides flanking the RT-LAMP target sites (Fig. 1B and Supplementary Table 1) immobilized on paramagnetic beads. This step reduces non-target nucleic acids and other unwanted components in the biological samples and also concentrates the viral RNA. Comparing the Ct-values of RT-qPCR targeting the *E* gene[17] after silica-based RNA-extraction with hybridization capture targeting the *E* gene for the same volume of gargle lavage suggests a capture efficiency of roughly 5% (Fig. 1D). To allow comparison to RT-qPCR we captured viral RNA with a

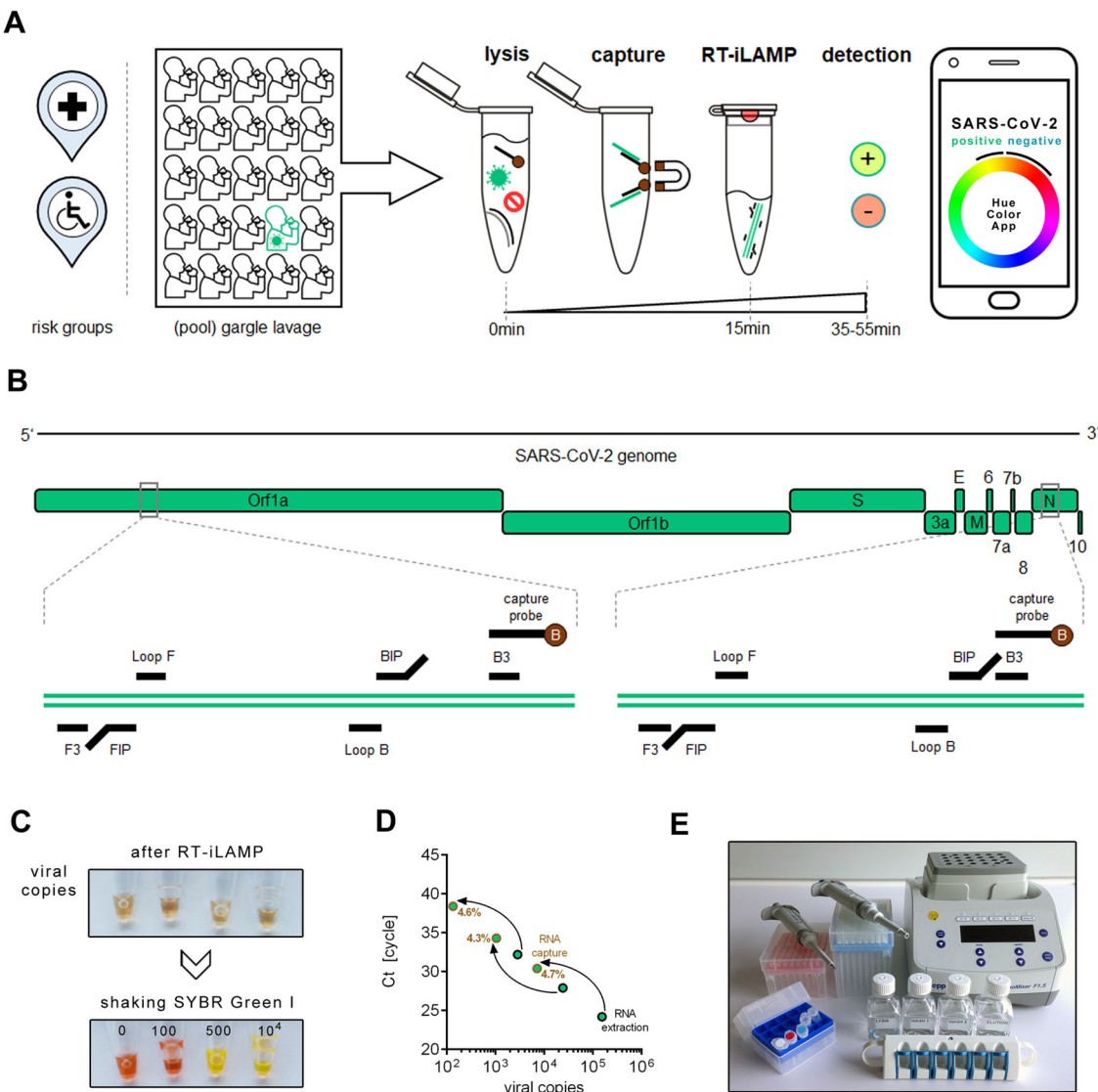

**Fig. 1 Cap-iLAMP to detect SARS-CoV-2. A** Workflow of Cap-iLAMP involves collecting and optionally pooling up to 26 gargle lavage samples, followed by combined lysis, target RNA enrichment, and improved LAMP (iLAMP). Color hue values can be obtained using any freely available "camera color picker" application on a smartphone. **B** Positions of primers and biotinylated capture oligonucleotides targeting the viral *Orf1a* and *N* gene on the SARS-CoV-2 genome. **C** Color change induced by mixing a drop of SYBR green I in the lid of the tube after iLAMP reaction with different input copy numbers of synthetic viral RNA. **D** Capture efficiency of three samples with different viral loads was estimated relative to automated silica-based RNA extraction based on copy number estimates for RT-qPCR assay targeting the SARS-CoV-2 *E* gene. Source data are provided as a Source Data file. **E** Equipment necessary for Cap-iLAMP: Pipettes and pipette tips, pre-mixed reagents (including iLAMP master mix and capture bead suspension), stable buffers (lysis/binding buffer, wash buffer, low salt buffer, elution buffer), a magnetic rack, and a thermoblock. A smartphone (not depicted) is recommended for hue color scoring.

biotinylated probe for the *E* gene and not for the *Orf1a* and *N* gene as used for RT-iLAMP. When RNA is concentrated from 500 µl gargle lavage to 25 µl final volume and 10 µl input volume is used in iLAMP, this results in a detection limit of 5–25 viral genome copies per µl of the sample before capture as the final Cap-iLAMP formulation detects 100–500 viral copies per reaction (Supplementary Fig. 6A, B).

The reagents required for the iLAMP reaction can be pre-mixed and freeze-thawed at least twice. Cap-iLAMP could be performed at point-of-care as only pipettes, a thermoblock, a magnetic rack, and a smartphone are needed (Fig. 1E). The price of two replicates experiments for a single test including capture and assays for the *Orf1a* and *N* gene, as well as a positive control and negative control, is ~30 € (Supplementary Table 2). Thus the reagent cost for testing a single individual in a pool of 26 individual gargle lavage samples is around 1€.

**Detection of SARS-CoV-2 in gargle lavage samples.** To validate the method on real samples, we used gargle lavage from hospital patients and nursing home inhabitants and employees. All samples had previously been tested by an RT-qPCR assay targeting the SARS-CoV-2 *E* gene[1]. Cap-iLAMP targeting the SARS-CoV-2 *Orf1a* gene or the *N* gene results in an orange/red and intense yellow color for SARS-CoV-2 RNA negative and positive samples, respectively (Supplementary Fig. 7A). We initially tested 12 individual gargle lavage samples employing only two wash steps after capture. Of these samples, six had previously been tested positive in the RT-qPCR assay (Supplementary Fig. 7B). Scoring the color of the iLAMP reactions using a smartphone "camera color picker" app shows that hues for SARS-CoV-2-negative and positive samples are clearly separated, below 26° for the former and above 37° for the latter, respectively (Supplementary Fig. 7C). All six negative samples were identified as negative. Of the six

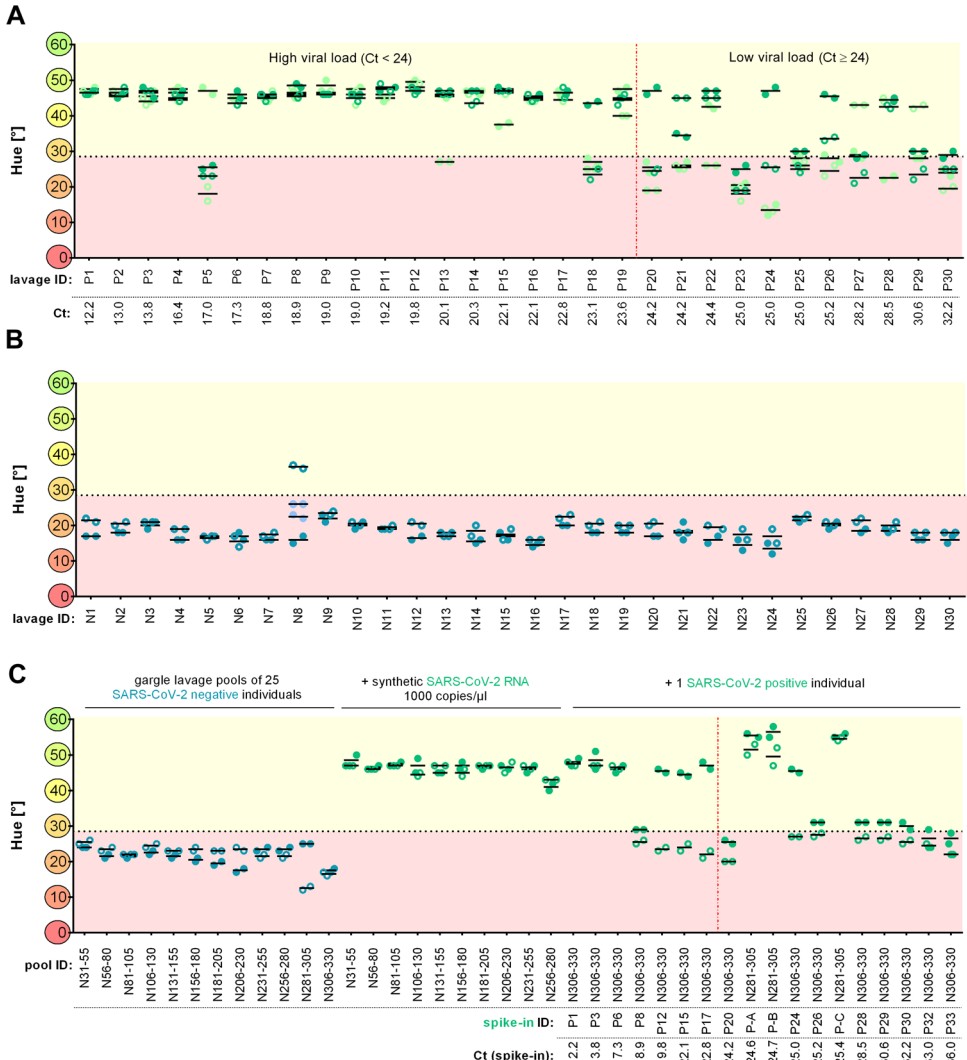

**Fig. 2 Detection of SARS-CoV-2 in gargle lavage samples.** Data obtained from healthy individuals are shown in blue while SARS-CoV-2-positive patient samples are depicted in green. Solid circles indicate values of the *Orf1a* assay while hollow circles denote values of the *N* gene assay. Assays with a hue >28.5° (dotted line) are considered positive. Repeated experiments are depicted in a light color and respective samples underwent an additional freeze/thaw cycle. **A** Hue of individual SARS-CoV-2-negative gargle lavage samples after Cap-iLAMP measured in duplicates. **B** Hue of individual SARS-CoV-2-positive gargle lavage samples after Cap-iLAMP measured in duplicates. Ct values of individual patient gargle lavage samples obtained via RT-qPCR assay targeting the SARS-CoV-2 *E* gene are stated. **C** Hue of gargle lavage pools after Cap-iLAMP measured in duplicates. The Ct values of spiked-in positive samples are stated. A vertical red dotted line indicates the separation between samples with high viral loads (Ct < 24) and low viral loads (Ct ≥ 24). Samples with Ct < 24 are likely highly infectious. Source data are provided as a Source Data file.

samples that were SARS-CoV-2-positive in the RT-qPCR assay, four were positive in the Cap-iLAMP assay while the remaining two were false negative. When one-twentieth of capture eluate volume was used they were positive, suggesting that some residual inhibition originating either from the biological sample or from lysis/binding buffer carryover exists in these extracts. Consequently, we employed three instead of two wash steps in all subsequent experiments. Because the hue was never above 28.5° in all subsequently tested negative samples in this study (n = 236), we consider assays with hue >28.5° as positive.

This Cap-iLAMP protocol was applied to test 30 individual gargle lavage samples in two replicate experiments. These samples had been previously tested positive with RT-qPCR and Ct values ranged from 12.2 to 32.2 (Fig. 2A and Supplementary Table 3). Nineteen of the samples had Ct values below 24 and are thus likely to be infectious[18] even if the ability of a person to infect others may vary as viral loads fluctuate over time[19] and often from sample to sample on the same occasion. Seventeen of the

19 samples tested positive in both assays targeting the *Orf1a* and the *N* gene, while the remaining two tested positive for one of the genes in one of the replicate experiments. Five of the 11 samples with Ct > 24 tested positive in assays for both genes in one of the replicates, and 10 tested positive for one of the genes in at least one of the replicates.

Next, we applied Cap-iLAMP to test individual gargle lavage samples previously found to be negative with RT-qPCR (Fig. 2B). Twenty-nine negative samples were correctly identified as negative in both Cap-iLAMP assays targeting the *Orf1a* and the *N* gene, while one showed amplification when targeting the *Orf1a* but not the *N* gene. Repeating the assays for this sample resulted in a negative assignment for both assays.

To assess the frequency of false-positive amplification under realistic conditions a blinded experimenter analyzed a set of 192 gargle lavage samples that contained a single positive sample (Ct 22) (Supplementary Fig. 8). The positive sample was correctly identified with both the *Orf1a* and *N* gene assays. Additionally, one sample

showed amplification in the assay targeting the *N* gene while another tested positive in the assay targeting the *Orf1a* gene. Suspiciously, these two samples were located in wells adjacent to the true positive sample in the plate containing the capture eluates, which could hint to experimental carryover. Repeating the assays for these samples resulted in correct negative assignment for both genes.

Next, we created 12 pools of 25 patient samples each, all of which had been tested negative in RT-qPCR assay and in the Cap-iLAMP assays for the *Orf1a* and the *N* gene (Fig. 2C). To determine if components of the pooled gargle lavage still inhibit the RT-LAMP reaction after capture, we took subsamples of negative pools and added 1000 copies/μl of artificial viral RNA before Cap-iLAMP. All pools were positive in both Cap-iLAMP assays, showing that there was no substantial inhibition in these extracts after capture.

To investigate whether it is possible to detect a single positive individual within a pool, we added different single positive patient samples with Ct values ranging from 12.2 to 36 to pools of uninfected individuals so that 1/26 (3.8%) of the final volume was composed of the infected sample (Supplementary Table 4). Six of the 18 gargle lavage pools tested positive in both assays targeting the *Orf1a* and the *N* gene, respectively, and nine additional pools tested positive for the assay targeting the *N* gene. The hues of the four samples were barely above the 28.5° threshold. This demonstrates that a single SARS-CoV-2-positive individual can be detected in a pool of 26 samples, albeit with some loss of sensitivity compared to individual sample testing.

**Detection of SARS-CoV-2 in heat-inactivated cell-culture supernatants.** Finally, we validated Cap-iLAMP against a set of seven reference samples of heat-inactivated cell-culture supernatants provided by the "INSTAND" reference institution of the German Medical Association (Duesseldorf, Germany) for round-robin testing. These contained various amounts of SARS-CoV-2 (ring 59, 61, 63, and 64), different coronaviruses (ring 60 HCoV OC43 and ring 65 HCoV 229E), or no virus (ring 62) and had previously been tested via RT-qPCR[1] (Fig. 3A). As shown in Fig. 3B, all four SARS-CoV-2-containing samples were correctly and consistently identified by both the *Orf1a* and *N* gene assays while all samples devoid of virus or containing a different coronavirus were correctly classified as negative.

**Diagnostic predictive power of Cap-iLAMP.** As described above, we tested gargle lavages from single individuals ($n = 221$) and pools of 25 healthy individuals ($n = 12$) from a sum of 521 SARS-CoV-2-negative hospital patients and nursing home inhabitants and employees, as well as round-robin test cell-culture supernatants void of SARS-CoV-2 ($n = 3$). Among those, there was no SARS-CoV-2-negative sample for which both Cap-iLAMP assays targeting the *Orf1a* or the *N* gene showed amplification. For three samples, one assay was positive, but subsequent analysis of those was negative for both assays.

We propose two approaches for diagnostic assignment (Supplementary Table 5). A conservative approach requires both assays targeting the *Orf1a* and the *N* gene, respectively, to be positive in at least one of two replicates in order to assign a sample as SARS-CoV-2-positive, resulting in a false-positive rate of 0% (0/236, 95% binomial confidence interval 0–1.3%) (Supplementary Table 6). The relaxed approach requires at least one assay to be positive to assign the sample as presumptive positive for SARS-CoV-2, resulting in a false-positive rate of 1.3% (3/236, 95% CI: 0.3–3.7%). For samples with high viral loads (Ct < 24), the conservative false negative rate is 9.1% (2/22, 95% CI: 1.1–29.2%), while the relaxed false negative rate is 0% (0/22, 95% binomial CI: 0–15.4%). Considering all SARS-CoV-2-positive samples together, the conservative false negative rate is 20% (7/35, 95% CI: 8.4–36.9%) and the relaxed false negative rate is 2.9% (1/35, 95% CI: 0.1–14.9%). In the pools of 26 gargle lavage samples, the conservative false negative rate is 66.7% (12/18, 95% CI: 41–86.7%) and the relaxed false negative rate is 16.7% (3/18, 95% CI: 1.4–34.7%). For the pools that contain gargle lavage from individuals with Ct < 24, the relaxed false negative rate is 0% (0/7, 95% CI 0–41%).

### Discussion

Cap-iLAMP overcomes problems associated with standard RT-LAMP pH-dependent colorimetric detection which is prone to false positives due to pH-variability in gargle lavages and off-target amplification. Gargle lavage cause no discomfort and is thus more suitable than pharyngeal and nasal swabs for the frequent and repeated screening of apparently healthy individuals[20]. They have the additional advantage that they do not expose the person collecting samples to any risk of infection.

In contrast to RT-qPCR approaches, no complicated equipment such as qPCR machines is required. Removal of inhibitors

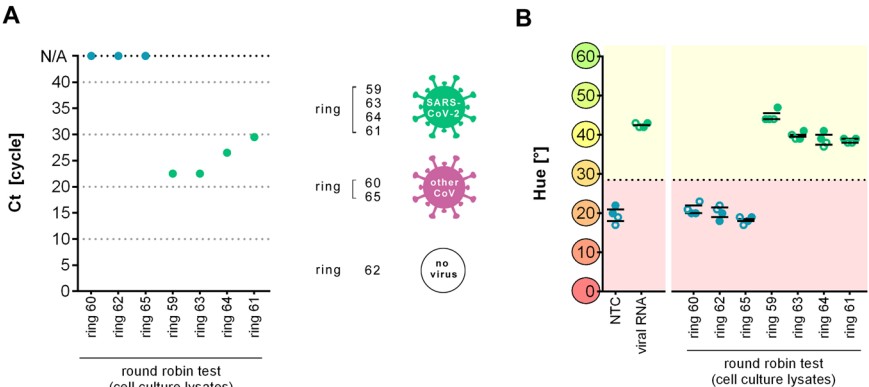

**Fig. 3 Detection of SARS-CoV-2 in cell-culture supernatants provided for assay validation. A** Ct values of individual cell-culture supernatant samples obtained via RT-qPCR assay targeting the SARS-CoV-2 *E* gene. **B** Hue of individual cell-culture supernatant samples after Cap-iLAMP measured in duplicates. Solid circles indicate values of the *Orf1a* assay while hollow circles denote values of the *N* gene assay. Data obtained from healthy individuals are shown in blue while SARS-CoV-2-positive patient samples are depicted in green. Assays with a hue >28.5° (dotted line) are considered positive. Source data are provided as a Source Data file.

and enrichment of target nucleic acids by bead capture, combined with a double assay of two SARS-CoV-2-specific target sites increase accuracy. As quality controls, we recommend water negative controls and sample-specific inhibition/positive controls containing the sample and synthetic viral RNA for each the *Orf1a* and the *N* gene assay. A scheme for evaluating the results is shown in Supplementary Table 5.

Due to the removal of non-target nucleic acids and the inclusion of LAMP enhancing enzymes in Cap-iLAMP, we drastically reduce false-positive results, which were reported for previous LAMP-based detection methods[21–23]. The inclusion of dye in the cap of tubes allows them to remain closed for detection after amplification thus reducing the risk of the contamination of reagent with amplification products. We quantified color as a single numeric hue value using a freely available smartphone app, which enables point-of-care detection without the ambiguity of an individual's perception of color. Yin et al.[15] already described the application of hue for hydroxynaptholblue scoring that, however, requires a custom-made "smart cup" reaction tube holder as well as an app that is not freely available. Table 1 compares key aspects of Cap-iLAMP to other published SARS-CoV-2 detection methods[6,11,12,17].

When testing individual samples we correctly identify 100 or 91% of positive samples with high viral loads (Ct < 24) with the relaxed (at least one assay targeting the *Orf1a* or the *N* gene positive) and the conservative assignments (both assays positive), respectively. Among all individual positive samples, we correctly identify 97.1% or 80%, respectively. This level of sensitivity is obviously not sufficient for regular diagnostic use. However, Cap-iLAMP could be useful for screening purposes to limit the spread of the infection, especially in settings where laboratory facilities and equipment are not available. The relaxed assignment is preferable when false positives are more acceptable than false negatives, e.g., when screening personnel or visitors to a nursing home or hospital. When compared to antigen tests which are currently used for screening in such settings, Cap-iLAMP (in its relaxed form) has similar specificity and higher sensitivity. In a meta-analysis of antigen tests, the average sensitivity of antigen tests was 56.2% (95% CI 29.5–79.8) and the average specificity 99.5% (95% CI: 98.1–99.9%)[24]. A comparison of sensitivity and specificity of different point-of-care tests for SARS-CoV-2 is shown in Supplementary Table 7.

We correctly identify 100% of pools that contain one sample with a high viral load (Ct < 24) and 83% of all positive pools with the relaxed criteria. This opens the prospect of rapid screening of large numbers of people for individuals who may be highly infectious in settings such as in schools, hospitals, airports, prisons, retirement homes, or border crossings, and allows daily testing of people at risk or working in critical infrastructure. Based on the relaxed false-positive detection rate, for every 2600 individuals (100 pools) tested, one pool would on average be falsely assigned as positive. Subsequent testing of individuals from positively tested pools might be assigned with the conservative approach to exclude false-positive assignment when infection incidence is low.

The Cap-iLAMP method can be readily applied to numerous human respiratory pathogens, plant pathogens, animal pathogens, food-borne diseases, viruses, protozoan parasites, and fungi, for which primer combinations have been developed[3]. These include Marburg virus disease, Lassa fever, Crimean-Congo hemorrhagic fever, Ebola virus disease, Middle East respiratory syndrome coronavirus, Nipah disease, Rift Valley fever, and Zika, that are prioritized by the WHO (as of July 2020) as they pose the greatest public health risk due to their epidemic potential and/or whether there is no or insufficient countermeasures[25–32]. For testing in remote areas master mixes

**Table 1 Comparison of SARS-CoV-2 detection methods.**

| | RNA extraction and RT-qPCR | DETECTR RT-LAMP/Cas12a | Direct sample to RT-LAMP | Cap-RT-iLAMP |
|---|---|---|---|---|
| Reference | Corman et al. | Broughton et al. | Dao Thi et al., Buck et al. | This study |
| Sample-to-result time | 4 h | 45 min | 30 min | 55 min |
| Reagent price | $$$ | $$ | $ | $ |
| Bulky equipment | Yes | No | No | No |
| Limit of detection | 1 copy | 10 copies | 100–500 copies | 100–500 copies (eq.-5–25 copies per µl sample liquid before capture) |
| Inhibitor removal | Yes | No | No | Yes |
| Sporadic false positives | No | No | Yes | No |
| Open-lid cross-contamination | No | Yes | No | No |
| Assay result detection | Fluorescence (FAM) | Colorimetric (FAM) | Colorimetric (pH-dye) | Colorimetric (cap SYBR Green I) |
| Helicase or PPase addable | Not applicable | Yes | No | Yes |
| Assay result scoring | Quantitative Ct value | Lateral flow | Subjective visual scoring | Smartphone → quantitative hue |

Ten important aspects of different published SARS-CoV-2 detection methods and Cap-iLAMP are compared.

for iLAMP could be lyophilized for transportation or long-term storage, as shown previously[33,34].

## Methods

**Sampling of gargle lavages.** 10 ml of sterile water was gargled for 10 s and spit back into a sterile scaled urine cup[1]. All individuals included in this study were asked for their voluntary assistance to participate and each individual gave written informed consent before entry into the study. The study was approved by the Ethics Committee of the Saxonian medical chamber (EK-allg-37/10–1). All procedures utilized in this study are in agreement with the 1975 Declaration of Helsinki. Analyzed anonymized gargle lavages from hospitalized COVID-19 patients and round-robin samples (INSTAND) were kept frozen below −80 °C for 1–2 months.

**In solution capture purification of SARS-CoV-2 nucleic acids.** Per reaction 20 µl Dynabeads MyOne Streptavidin C1 bead suspension (ThermoFisher Scientific, #650.02) are pelleted using a magnetic rack and washed twice with 500 µl combined 1× lysis/binding buffer (LysBB: 100 mM Tris-HCl, pH 7.5, 500 mM LiCl, 0.5% LiDS, 1 mM EDTA, 5 mM DTT). Washed beads are then resuspended in 100 µl 1× LysBB before the addition of 20pmol CV2_btn and 20pmol CV16_btn (Supplementary Table 1) (or 10pmol E_Sarbeco_R2_btn for the RT-qPCR assay) and rotation for 10 min at room temperature. Beads are pelleted and the supernatant is discarded before resuspension in 200 µl alkaline wash solution (125 mM NaOH, 0.1% (v/v) Tween-20) and 5 min incubation at room temperature. Beads are pelleted with a magnetic rack and supernatant is discarded. Beads are then washed once with 200 µl 1× LysBB before resuspension in 500 µl 2× LysBB. Prepared bead suspension can be stored at 5 °C or used directly.

To 500 µl ready-made bead suspension an equal volume of patient gargle lavage (sample), water (negative control), or sample with 500k copies of artificial SARS-CoV-2 RNA (Twist Biosciences, #102024) (positive and inhibition control) is added and mixed by pipetting. Copy numbers were reported according to specifications by the commercial provider. The reaction is incubated at 55 °C for 10 min and then placed on a magnetic rack. The supernatant is discarded and the beads are thoroughly resuspended in 200 µl wash buffer (20 mM Tris-HCl, pH 7.5, 500 mM LiCl, 1 mM EDTA). Beads are pelleted and the supernatant is discarded, repeat this step once for individual or inhibited samples. Wash beads once with 200 µl low salt buffer (20 mM Tris-HCl, pH 7.5, 200 mM LiCl, 1 mM EDTA) and discard the supernatant. Beads are resuspended in 25 µl elution buffer (20 mM Tris-HCl, pH 7.5, 1 mM EDTA) and RNA is eluted by heating to 60 °C for 2 min. Transfer the supernatant to a fresh 0.2 ml strip tube or use directly as input for RT-iLAMP or RT-qPCR.

**RT-iLAMP.** For each sample, two assays targeting either the *Orf1a* or *N* gene are performed. Primer mixes for both assays are prepared in bulk for quick reaction assembly (Supplementary Table 3). 20 µl of RT-iLAMP master mix containing either the *Orf1a* (CV1-6) or *N* gene (CV15-20) primer sets and 10 µl of capture eluate are combined to obtain a final reaction volume of 30 µl (1× WarmStart® Colorimetric LAMP Master Mix, NEB, Ipswich, MA, USA, #M1800L), 1 mM ATP, 1 µM SYTO 9, 1× primer mix, 0.1 ng/µl Tte UvrD Helicase (NEB, #M1202S), 0.05 U/µl thermostable inorganic pyrophosphatase (NEB, #M0296L), 0.4 U/µl Protector RNase inhibitor (Sigma Aldrich, St. Louis, MI, USA, # 03335402001) (Supplementary Tables 8 and 9). For color reactions, 0.5 µl SYBR green I is applied to the lid of the tube without getting into contact with the reaction liquid. For transport or preparation of large numbers of reactions, it is also possible to immobilize the dye by drying for 15 min at 70 °C in an oven. The reaction is incubated at 65 °C for 40 min. If SYBR green I was applied to the cap of the tube, the reaction is then stopped by shaking and can be evaluated by eye or with any "camera color-picker" app on a smartphone within a minute. Photographs were taken using Redmi 7 model M1810F6LG (and iPhone 5 model 1429 for Supplementary Fig. 3). We used the "Palette Cam" app (Alexander Mathers, App Store) to score red, green, and blue (RGB) values and extracted the hue. This can be easily done using a web tool (e.g., https://www.rapidtables.com/convert/color/rgb-to-hsv.html) or in Microsoft Excel (=ROUNDDOWN(IF(180/PI()*ATAN2(2*$R-G-B$,SQRT(3)*(*G-B*)) < 0,180/PI()*ATAN2(2*$R-G-B$,SQRT(3)* (*G-B*)) + 360,180/PI()*ATAN2(2*$R-G-B$,SQRT(3)*(*G-B*))),0)). Hue can also be scored directly when using e.g., "Color Grab (color detection)" app for android or "Aurora" app for iOS.

A step-by-step protocol is available on Protocol Exchange (DOI: 10.21203/rs.3.pex-1344/v1)[35].

**Reporting summary.** Further information on research design is available in the Nature Research Reporting Summary linked to this article.

## Data availability

Data are available on request from the authors. The images of colorimetric assays are available on the Dryad repository (https://datadryad.org/stash/dataset/doi:10.5061/dryad.2rbnzs7mk). Source data are provided with this paper.

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

## Acknowledgements
We thank the management of Diakonische Dienste Leipzig, the personnel and residents of Altenpflegeheim Emmaus in Leipzig, and the personnel of the Department of Laboratory Medicine at Hospital St Georg Leipzig for their cooperation. Funding was provided by the Max Planck Society, the NOMIS foundation, and the ImmunoDeficiencyCenter Leipzig.

## Author contributions
S.R., L.B., M.M., and S.P. conceived the idea. L.B., S.R., M.M., T.M, S.P., and S.B. planned experiments. L.B. and S.R. performed experiments and analyzed data. S.B. and O.N. provided resources and helped with validation. S.R., L.B., and S.P. wrote the paper with input from all authors.

## Funding

## Competing interests
The authors declare no competing interests.
