## [Peer Review File · Nature Communications]

Reviewers' Comments:

Reviewer #1:

Remarks to the Author:

Reviewer report for "Rapid, reliable, and cheap point-of-care bulk testing for SARS-CoV-2 by combining hybridization capture with improved colorimetric LAMP (Cap-iLAMP)" by Bokelmann et al.

SUMMARY:

In this manuscript Bokelmann et al presents data on a modified version of iLAMP (improved loop-mediated isothermal amplification), "Cap-iLAMP", in which they added an RNA purification (capture) step. Amplified products were detected using SYBR Green I, either using a conventional qPCR machine (read-time assay) or by plate imaging (end point) and classifying light intensity and hue in wells using previously available software. The authors tested the method using synthetic SARS-CoV-2 RNA and on gargle lavage samples from patients diagnosed for COVID-19 by clinical RT-PCR.

The ambition to develop rapid, affordable and accurate COVID-19 testing methods is of course great. Unfortunately, I find the manuscript of limited value as currently presented and that it does not add enough clarity on multiple point. Rigorous evaluation of the method is lacking and based on the current data I think the method is being somewhat oversold. Methodologically the main difference from the previous iLAMP method is the addition of an RNA capture step (bead purification), reasonably making the method considerably more work intensive and moreover reliant on specifically designed target oligos for the magnetic capture of RNA targets. To what degree the RNA capture improves iLAMP is not systematically investigated. As presented, the method has a high false negative rate (~33%) when tested on individual samples and this was only evaluated on 6 + 6 positive and negative samples. Testing across patient samples of high to low viral load is not performed, which would be needed to determine the method's sensitivity. In further tests of pooled negative samples (n pools= 11) spiked with synthetic SARS-CoV-2 RNA, only high input load of RNA was tested (1000 copies/ul, i.e. 10,000 copies in the reaction, which is a lot). Finally, when performing testing on pools of 25 negative patient samples spiked with one SARS-CoV-2 positive patient sample, the spiked sample had been selected among those with the very highest viral load, which clearly does not reflect the random pooling of in a real situation. Until systematic and unbiased testing has been performed, I find the study not ready for publication.

MAJOR POINTS:

[1a] Results, page 3: How was the image acquisition made? Using a mobile phone or a plate reader? This vital information is lacking both in the Results and the Methods. Further, what model of phone or plate reader? I assume that different mobile phones would perform very differently due to the camera hardware. Also, include the used equipment (phone or plate reader) in Figure 1E ("Necessary equipment").

[1b] Again, in Results, page 4: "The reagents required for the iLAMP reaction can be pre-mixed and freeze-thawed at least twice. Cap-iLAMP can be done at point-of-care as no bulky equipment and only pipettes, a thermoblock, and a magnetic rack are needed (Figure 1E)." How about the equipment for image acquisition?

[2] Results, page 5: "Of the six samples that were SARS-CoV-2 positive in the RT-qPCR assay, four were positive in the Cap-iLAMP assay while the remaining two were false negative." This is a 33% false negative rate on individual sample level, which is huge and unacceptable for any medical test. Later, on the same page, the authors state that potential inhibitors present in rare individual samples might be the cause of the dropout ("...potentially because inhibition is rare and is diluted out in the pool"). But based on the few individual samples tested (n=6, and 2/6 dropouts) it does not seem rare at all but occurred in 2/6 cases. Testing across patient samples of high to low viral load was not performed, which would be needed to determine the method's

sensitivity.

[3] Results, page 5: The authors used 10 (or 11? – the text is ambiguous on this, see Minor points) pools of 25 negative patient samples each and spiked with 1000 copies/ul of synthetic SARS-CoV-2 RNA (Twist, commercial provider). The methodological description of this experiment is sparse, but from what I understand this would equate 10,000 purified genomic copies in the reaction (10ul reaction input). This is a rather high viral load. To get some idea about sensitivity it would be good to see the results using steps of less RNA input (for example 10000; 5000; 1000; 500; 100; 10; 0). This should be a simple experiment to perform given that spiked synthetic RNA is used.

[4] Results, page 5: "To investigate whether it is possible to detect a single infectious individual in a pool, we added different single positive patient samples (Ct < 26) to three pools of healthy individuals so that 1/26 (3.8%) of the final volume was composed of the infected sample." Why did you choose to spike the pools with samples of the very highest viral load (Ct < 26; range: 24.6 - 32.7)? This experiment is biased in that it does not represent spiking of a random positive samples, having Ct range: 24.6 - 32.7. The experiment should be replicated using spike of positive samples across the Ct range. Three positive sample pools is a very low number.

[5] A major drawback with the pooling method is that once a pool has been called as positive one would need to go back to the original samples to re-run individual tests. One would then face the problem with inhibition and ~33% false negatives (see point 2). This would also delay the diagnosis. As the study focuses on tests of pooled samples this problem should be discussed.

[6] I miss a discussion of the lack of an internal sample quality control in this method, as established clinical RT-PCR diagnostics always employ an internal target to test sample integrity.

[7] Methods and data availability: The Methods section lacks crucial information, for example information on image acquisition and processing. A link to a repository for the original data and computational code is missing.

MINOR POINTS:

[8] Abstract, page 1: Please provide the cost calculation for the stated ~1 Euro in the Methods and a mention in the Results.

[9] Introduction, page 2: When discussing RT-PCR and the potential benefit of LAMP assays to replace RT-PCR the authors describe RT-PCR methods as requiring trained personnel. This must surely be the case also for Cap-iLAMP (requiring bead purification etc.) and should not be phrased as something contrasting between the methods.

[9] Introduction, page 2: It would be fair to mention that sensitive RNA-extraction-free SARS-CoV-2 RT-PCR-based tests are now established.

[10] Introduction, page 4: "We compare the method sensitivity to standard extraction RT-qPCR protocols and validate it on 287 gargle lavage samples from a hospital..." This is inaccurate. The method's sensitivity as defined by standard terminology in medical testing [accuracy, sensitivity, specificity] has not been determined nor systematically compared with RT-qPCR. The best way to do this would have been to test the 287 samples individually using Cap-iLAMP and calculate accuracy, sensitivity and specificity using RT-qPCR as comparator.

[11] Results, page 3: It is stated that a spike of 500 copies of synthetic RNA is detected after 25-30 minutes of incubation. How do you determine the RNA copy number? Stated copy number of commercial RNA is known to often be inaccurate. If you did not experimentally quantify RNA copy numbers, you should state in the Methods that copy numbers were taken as described by the commercial provider but not validated. Commercial RNA should be tested to be DNA-template free

by excluding reverse transcriptase, as DNA contamination is a common issue in synthetic RNA.

[12] Results, page 3: "We found that nasopharyngeal swab eluates tend to be more acidic than gargle lavage samples and that adding gargle lavage directly to a LAMP reaction at a final concentration of 5% leads to false positive results in 4.7% of cases even before the isothermal incubation (Supp. Figure 2A and B)." It is stated that nasopharyngeal swab eluates tend to be more acidic than gargle lavage samples – where in the paper is this shown or where is the reference demonstrating this?

[13] Results: "We therefore employed a rapid (15min) bead-capture enrichment purification akin to mRNA isolation, using two oligonucleotides flanking the RT-LAMP target sites (Figure 1B, Supp. Table 1) immobilized on paramagnetic beads." I would not describe this as particularly rapid, but normal for an RNA purification step. Additionally, the capture requires the synthesis and preparation of specific oligo-mag conjugates. The addition of the purification step in terms of time as well as capture-bead production in contrast to previous iLAMP procedures should be mentioned in the Discussion.

[14] Results, page 5: "To investigate whether it is possible to detect single infected individuals in pools of gargle lavage samples, we created eleven pools of 25 patient samples each, all of which had tested negative in RT-qPCR assay and in the Cap-iLAMP assays for the Orf1a and the N gene (Figure 2D). In order to determine if components of the pooled gargle lavage still inhibit the RT-LAMP reaction after capture, we took subsamples of the 10 negative pools and added". What happened to the 11th sample pool? In Fig. 2D 11 negative pools are plotted but only 10 of them plotted as spiked. If one spiked was skipped, that is fine, but if so, clarify it in the legend.

Reviewer #2:

Remarks to the Author:

Bokelmann et al. present a workflow to detect SARS-CoV-2 infections in gargle lavage samples using LAMP. Over the last few months, quite a few papers have appeared that present approaches to detect SARS-CoV-2 in a manner that is simple, cheap and fast, and use of the LAMP assay is a common ingredient there. Given the urgency of this task, this flood of papers is certainly a good thing, as we are really getting to working solutions this way.

What do Bekelmann et al. add to the already existing suggestions? Their main new idea is to use hybridization capture: in order to isolate the RNA from the sample, magnetic beads are typically employed which adsorb RNA of any kind. The authors now adsorb oligos complementary to a locus on the viral genome close to the primer binding sites, in order to specifically enrich viral RNA, in the expectation that this will increase specificity. They also "mix and match" a number of other ideas, which have been proposed before but not in this combination, to improve LAMP-based detection further: they confirm an earlier report that addition of a certain helicase strongly suppresses the unspecific amplification that is considered a major weakness of LAMP. Also, they suggest to add the dye to the inside of the cap of the tube where it remains during the reaction so that one can mix it in later without having to open the tube.

The crucial question for LAMP assays is that of specificity. Given that even in hot spots, prevalence is typically only around 1%, we need a specificity above 99% in order to avoid raising more false than true alarms. The authors have tested negative 25 gargle-lavage pools without getting a false positive. The 95% binomial confidence interval for 25/25 is (using Wilson's method) 87-100%. In other words: with only 25 negative samples, it is hard to say that the assay has sufficient specificity.

To really be able to say (with 95% confidence) that the assay has a specificity over, say, 98%, one would need to see no false positive after testing at least ~190 negative samples.

This is, of course, quite an effort. However, without this, the authors' claim that they have better specificity than other recently published approaches cannot be made. This, of course, is not necessary to publish justification, but the text should somehow acknowledge that the available sample size is not sufficient to make too strong claims about specificity (or claims that there are "no" sporadic false positives, as Table 1 claims).

Put is in another way: giving binomial confidence intervals is really important whenever one makes claims about *not* having observed some outcome after having made n tries (as otherwise, papers with smaller will look stronger than those with large n).

Similar issues arise for other claims in Table 1: Does the addition of helicase really improve specificity? Supplementary Figure 3A makes a strong case for this claim, but the extent of this improvement in practice is hard to estimate without very many samples.

The same goes for the core result: would a conventional magnetic bead extraction really result in more false positives than the hybridization capture suggested here? (Or have I misunderstood and the advantage rather lies in simplicity? The authors perform only one washing step, and, I believe, usual magnetic bead extraction protocols use several.)

One way to clarify this might be to perform RT-qPCR before and after bead purification, as done for Figure 1D, but do this twice: once with beads with specific oligos and once with normal beads.

Minor issues:

- I'm not familiar with SybrGreen's orange-to-yellow colour change: is this caused by the same conformation change as the well known change to become fluorescent? Is it really due to intercalation only and independent of pH? A reference to a paper describing this might be helpful.

- The idea to put the dye into the tube cap sounded familiar, so I searched and found that it was discussed in a few older paper as "closed-tube LAMP" (e.g., <https://doi.org/10.1007/s00705-018-3706-0>). maybe the authors could try to find the original first reference to the idea and cite it?

- It's nice that the same two colours are used for negative and positive samples throughout, but I find the chosen bluish green and greenish blue very hard to distinguish.

- Figure 2F is too busy. I cannot see anything with all these lines. Maybe it could be split into two panels?

- Figure 2C: the bracket should be explained in the figure caption, not only in the main text.

- The authors claim that the use of a colour picking app gives absolute and reproducible results independent of subjective viewing preferences. I agree but worry about mobile phone's automatic white balancing functionality. Could the authors please check by measuring the hue of the same sample under different lighting conditions (e.g., fluorescent light from a standard ceiling light versus sunlight outside on a bright day). They may need to caution the user to always use the same lighting conditions.

- Figure 1D is hard to read: It would be better to use one axis for the CT value before and the other for the CT value after purification. The viral loads could then be shown as secondary axes.

Overall, I consider this submission to be a quite valuable contribution to the current efforts to find a good and simple test, although the text needs to be worded a bit more carefully at some places, in order to avoid overstating claims.

Response letter

We want to thank the reviewers for their thorough evaluation of the manuscript and constructive comments and suggestions. We conducted a major revision of our work and hope to have addressed all concerns in a satisfactory way.

Reviewer 1

Reviewer #1 (Remarks to the Author):

Reviewer report for "Rapid, reliable, and cheap point-of-care bulk testing for SARS-CoV-2 by combining hybridization capture with improved colorimetric LAMP (Cap-iLAMP)" by Bokelmann et al.

SUMMARY:

In this manuscript Bokelmann et al presents data on a modified version of iLAMP (improved loop-mediated isothermal amplification), "Cap-iLAMP", in which they added an RNA purification (capture) step. Amplified products were detected using SYBR Green I, either using a conventional qPCR machine (read-time assay) or by plate imaging (end point) and classifying light intensity and hue in wells using previously available software. The authors tested the method using synthetic SARS-CoV-2 RNA and on gargle lavage samples from patients diagnosed for COVID-19 by clinical RT-PCR.

The ambition to develop rapid, affordable and accurate COVID-19 testing methods is of course great. Unfortunately, I find the manuscript of limited value as currently presented and that it does not add enough clarity on multiple point. Rigorous evaluation of the method is lacking and based on the current data I think the method is being somewhat oversold. Methodologically the main difference from the previous iLAMP method is the addition of an RNA capture step (bead purification), reasonably making the method considerably more work intensive and moreover reliant on specifically designed target oligos for the magnetic capture of RNA targets. To what degree the RNA capture improves iLAMP is not systematically investigated. As presented, the method has a high false negative rate (~33%) when tested on individual samples and this was only evaluated on 6 + 6 positive and negative samples. Testing across patient samples of high to low viral load is not performed, which would be needed to determine the method's sensitivity. In further tests of pooled negative samples (n pools= 11) spiked with synthetic SARS-CoV-2 RNA, only high input load of RNA was tested (1000 copies/ul, i.e. 10,000 copies in the reaction, which is a lot). Finally, when performing testing on pools of 25 negative patient samples spiked with one SARS-CoV-2 positive patient sample, the spiked sample had been selected among those with the very highest viral load, which clearly does not reflect the random pooling of in a real situation. Until systematic and unbiased testing has been performed, I find the study not ready for publication.

MAJOR POINTS:

1.

[1a] Results, page 3: How was the image acquisition made? Using a mobile phone or a plate reader? This vital information is lacking both in the Results and the Methods. Further, what model of phone or plate reader? I assume that different mobile phones would perform very differently due to the camera hardware. Also, include the used equipment (phone or plate reader) in Figure 1E ("Necessary equipment"). [1b] Again, in Results, page 4: "The reagents required for the iLAMP reaction can be pre-mixed and freeze-thawed at least twice. Cap-iLAMP can be done at point-of-care as no bulky equipment and only pipettes, a thermoblock, and a magnetic rack are needed (Figure 1E)." How about the equipment for image acquisition?

This is an excellent point. We use a smartphone with a color-picker app and now are more specific in describing this in the results section:

'We used the 'Palette Cam' app (Alexander Mathers, App Store) for extracting RGB values before conversion to hue. We tested the influence of two smartphone models (Redmi 7 and iPhone 5), as well as two different light sources (daylight and fluorescent tube light) on the obtained hue value. While both smartphones result in comparable hue values, daylight resulted in a clearer separation between negative and positive samples, than fluorescent tube light (Supp. Fig. 3 A and B). We thus acquired all images using daylight.'

We also added the smartphone to the list of required equipment in the results section on page 4 and included a sentence in the legend of Fig. 1E: 'A smartphone (not depicted) is recommended for hue color scoring.'

A plate reader was used to record the fluorescence of the SYTO-9 dye intercalating into the double-stranded DNA reaction products during technical development of the iLAMP. We clarified this in the Materials and Methods section and added the exact model of plate reader used: 'If amplification curves should be recorded on a qPCR machine (Bio Rad, CFX96 Real Time System), no SYBR green I is needed.'

2. Results, page 5: "Of the six samples that were SARS-CoV-2 positive in the RT-qPCR assay, four were positive in the Cap-iLAMP assay while the remaining two were false negative." This is a 33% false negative rate on individual sample level, which is huge and unacceptable for any medical test. Later, on the same page, the authors state that potential inhibitors present in rare individual samples might be the cause of the dropout ("...potentially because inhibition is rare and is diluted out in the pool"). But based on the few individual samples tested (n=6, and 2/6 dropouts) it does not seem rare at all but occurred in 2/6 cases. Testing across patient samples of high to low viral load was not performed, which would be needed to determine the method's sensitivity.

Indeed, the results shown in that figure (now Supp. Fig. 7) constitute an intermediate step in the development of the method as only two wash steps after capture were performed and residual inhibition sometimes caused false negatives. The final method which we had already used for the pools involved three post-capture washes to eliminate this problem. We therefore moved the figure into the supplement. Also, to address the question of sensitivity, we now tested 30 additional individual patient samples with a range of viral loads (Ct values ranging from 12.2 to 32.2) (Fig. 2A).

3. Results, page 5: The authors used 10 (or 11? – the text is ambiguous on this, see Minor points) pools of 25 negative patient samples each and spiked with 1000 copies/ul of synthetic SARS-CoV-2 RNA (Twist, commercial provider). The methodological description of this experiment is sparse, but from what I understand this would equate 10,000 purified genomic copies in the reaction (10ul reaction input). This is a rather high viral load. To get some idea about sensitivity it would be good to see the results using steps of less RNA input (for example 10000; 5000; 1000; 500; 100; 10; 0). This should be a simple experiment to perform given that spiked synthetic RNA is used.

This is a valid point; we now tested a range of RNA input amounts (1e5, 1e4, 1000, 500, 100, 10, 0 viral genome copies) with the final formulation of the iLAMP to assess the sensitivity (Supp. Fig. 6) and updated the relevant section in the main text:

'When RNA is concentrated from 500µl gargle lavage to 25µl final volume and 10µl input volume is used in iLAMP, this results in a detection limit of 5-25 viral genome copies per µl of sample before capture as the final Cap-iLAMP formulation detects 100-500 viral copies per reaction (Supp. Fig. 6A and B).'

4. Results, page 5: "To investigate whether it is possible to detect a single infectious individual in a pool, we added different single positive patient samples (Ct < 26) to three pools of healthy individuals so that 1/26 (3.8%) of the final volume was composed of the infected sample." Why did you choose to spike the pools with samples of the very highest viral load (Ct < 26; range: 24.6 - 32.7)? This experiment is biased in that it does not represent spiking of a random positive samples, having Ct range: 24.6 - 32.7. The experiment should be replicated using spike of positive samples across the Ct range. Three positive sample pools is a very low number.

We now tested 15 additional pools of 26 samples containing one positive sample with a range of different Ct values (12.2 to 36) (see Fig. 2C), giving us a more complete picture of the limits of the method.

5. A major drawback with the pooling method is that once a pool has been called as positive one would need to go back to the original samples to re-run individual tests. One would then face the problem with inhibition and ~33% false negatives (see point 2). This would also delay the diagnosis. As the study focuses on tests of pooled samples this problem should be discussed.

It is certainly true that once a pool has been found to be positive one has to run single-sample tests to identify the infected individuals. However, in most situations only a few percent of people are positive so it is likely that many pools will be negative, thus making it possible to quickly zoom in on the few infected individuals and saving time.

6. I miss a discussion of the lack of an internal sample quality control in this method, as established clinical RT-PCR diagnostics always employ an internal target to test sample integrity.

To test for sample-specific inhibition, we recommend an inhibition control with spiked-in artificial viral RNA. Unlike in PCR where multiple primer pairs and probes with different dyes can easily be combined in one reaction to generate an internal control signal, standard LAMP as employed here relies on detection of double-stranded DNA and does not easily allow for same-tube controls.

In the discussion we write: 'As quality controls we recommend dedicated water negative controls and sample-specific inhibition/positive controls containing sample and synthetic viral RNA for each the Orf1a and the N gene assay. A scheme for evaluating the results is shown in Supp. Table 5.'

7. Methods and data availability: The Methods section lacks crucial information, for example information on image acquisition and processing. A link to a repository for the original data and computational code is missing.

We added the relevant information to the end of the Methods section. We have uploaded the colorimetric assay images to the Dryad repository (doi:10.5061/dryad.2rbnzs7mk) and attached a Data availability statement in the Methods section. The repository data can be accessed prior to publication using [this temporary link: https://datadryad.org/stash/share/7qvSoBEyGKuDgiWFGi1S7oPMpTpNX9sm4C52O5rSRDY](https://datadryad.org/stash/share/7qvSoBEyGKuDgiWFGi1S7oPMpTpNX9sm4C52O5rSRDY).

MINOR POINTS:

8. Abstract, page 1: Please provide the cost calculation for the stated ~1 Euro in the Methods and a mention in the Results.

We added a cost calculation to the Supplementary material (Supplementary Table 2) and now write in the results section 'The price of two replicate experiments for a single test including assays for the Orf1a and N gene, as well as a positive control and a negative control, is approximately 30€ (Supplementary Table 2). Thus the cost for testing a single individual in a pool of 26 individual gargle lavage samples is around 1€.'

9 and 10. Introduction, page 2: When discussing RT-PCR and the potential benefit of LAMP assays to replace RT-PCR the authors describe RT-PCR methods as requiring trained personnel. This must surely be the case also for Cap-iLAMP (requiring bead purification etc.) and should not be phrased as something contrasting between the methods. It would be fair to mention that sensitive RNA-extraction-free SARS-CoV-2 RT-PCR-based tests are now established.

We have rephrased the sentence in question: 'However, its need for expensive bulky instrumentation and shortages of resources for RNA purification has spurred the search for viable alternatives even though sensitive RNA-extraction-free SARS-CoV-2 RT-qPCR-based tests are now established (Maricic et al. 2020).'

11. Introduction, page 4: "We compare the method sensitivity to standard extraction RT-qPCR protocols and validate it on 287 gargle lavage samples from a hospital..." This is inaccurate. The method's sensitivity as defined by standard terminology in medical testing [accuracy, sensitivity,

specificity] has not been determined nor systematically compared with RT-qPCR. The best way to do this would have been to test the 287 samples individually using Cap-iLAMP and calculate accuracy, sensitivity and specificity using RT-qPCR as comparator.

We changed the sentence in question to ‘We compared our method to standard extraction RT-qPCR protocols in a diagnostic lab and validate its performance on 570 gargle lavage samples from a hospital, a nursing home previously affected by COVID-19, and round robin samples from a reference institution of the German Medical Association.’

We have now tested a total of 236 negative samples and 35 SARS-CoV-2 positive samples as determined via RNA extraction followed by RT-qPCR (Ct values ranging from 12.2 to 36). If requiring just one assay (targeting either the Orf1a or N gene) to be positive in order to diagnose a sample as positive, the false positive rate is 1.3% (95% binomial CI 0.3 – 3.7%) and the false negative rate is 2.9% (95% binomial CI 0.1 - 14.9%).

12. Results, page 3: It is stated that a spike of 500 copies of synthetic RNA is detected after 25-30 minutes of incubation. How do you determine the RNA copy number? Stated copy number of commercial RNA is known to often be inaccurate. If you did not experimentally quantify RNA copy numbers, you should state in the Methods that copy numbers were taken as described by the commercial provider but not validated. Commercial RNA should be tested to be DNA-template free by excluding reverse transcriptase, as DNA contamination is a common issue in synthetic RNA.

We added a clarification to the Methods section: ‘Copy numbers were reported according to specifications by the commercial provider’.

13. Results, page 3: “We found that nasopharyngeal swab eluates tend to be more acidic than gargle lavage samples and that adding gargle lavage directly to a LAMP reaction at a final concentration of 5% leads to false positive results in 4.7% of cases even before the isothermal incubation (Supp. Figure 2A and B).” It is stated that nasopharyngeal swab eluates tend to be more acidic than gargle lavage samples – where in the paper is this shown or where is the reference demonstrating this?

It is true that we did not measure the pH of the gargle lavages or the swab eluates by any other means than adding them to the pH dye containing LAMP reaction mix. We clearly observe a stronger yellow shift in swab eluates than in gargle lavage samples as can be seen from the triangles in Supp. Fig. 2A. While there might be alternative explanations, a lower pH in the former sample type seems most plausible to us.

We now write: ‘We found that nasopharyngeal swab eluates tend to be more acidic than gargle lavage samples based on pH-dye color change in the LAMP mix and that adding gargle lavage directly to a LAMP reaction at a final concentration of 5% leads to false positive results in 4.7% of cases even before the isothermal incubation (Supp. Figure 2A and B).’

14. Results: “We therefore employed a rapid (15min) bead-capture enrichment purification akin to mRNA isolation, using two oligonucleotides flanking the RT-LAMP target sites (Figure 1B, Supp. Table 1) immobilized on paramagnetic beads.” I would not describe this as particularly rapid, but normal for an RNA purification step. Additionally, the capture requires the synthesis and preparation of specific oligo-mag conjugates. The addition of the purification step in terms of time as well as capture-bead production in contrast to previous iLAMP procedures should be mentioned in the Discussion.

Our point of reference was the automated RNA extraction used in combination with the RT-qPCR protocol (MagNA Pure 24 System, Roche) that typically takes 70 minutes for a total of 24 samples (see https://lifescience.roche.com/en_de/brands/magnapure.html). In this context, we deemed it appropriate to speak of a rapid 15min capture especially when pooling 26 individual samples and the per sample extraction time is far superior to automated silica-based extraction. As beads can be prepared in bulk and stored we did not include the time for bead preparation in our time estimate. The purification steps (lysis, capture and washes) are included in the time estimate (see Fig. 1A).

15. Results, page 5: “To investigate whether it is possible to detect single infected individuals in pools of gargle lavage samples, we created eleven pools of 25 patient samples each, all of which had tested negative in RT-qPCR assay and in the Cap-iLAMP assays for the Orf1a and the N gene (Figure 2D).

In order to determine if components of the pooled gargle lavage still inhibit the RT-LAMP reaction after capture, we took subsamples of the 10 negative pools and added". What happened to the 11th sample pool? In Fig. 2D 11 negative pools are plotted but only 10 of them plotted as spiked. If one spiked was skipped, that is fine, but if so, clarify it in the legend.

Indeed, of the 11 (now 12) pools in Fig. 2D (now Fig. 2C) only 10 received the SARS-CoV-2 RNA spike-in to show that these pools are not inhibited. The 11th and 12th pool were spiked in with real SARS-CoV-2 gargle lavage. They are also not inhibited, because spiked-in positive gargle lavage results in positive Orf1a and N gene assays for both of them. To make this clearer, we added the spike-in ID and Ct value to the pool-ID below the X-axis.

Reviewer #2 (Remarks to the Author):

Bokelmann et al. present a workflow to detect SARS-CoV-2 infections in gargle lavage samples using LAMP. Over the last few months, quite a few papers have appeared that present approaches to detects SARS-CoV-2 in a manner that is simple, cheap and fast, and use of the LAMP assay is a common ingredient there. Given the urgency of this task, this flood of papers is certainly a good thing, as we are really getting to working solutions this way.

What do Bokelmann et al. add to the already existing suggestions? Their main new idea is to use hybridization capture: in order to isolate the RNA from the sample, magnetic beads are typically employed which adsorb RNA of any kind. The authors now adsorb oligos complementary to a locus on the viral genome close to the primer binding sites, in order to specifically enrich viral RNA, in the expectation that this will increase specificity. They also "mix and match" a number of other ideas, which have been proposed before but not in this combination, to improve LAMP-based detection further: they confirm an earlier report that addition of a certain helicase strongly suppresses the unspecific amplification that is considered a major weakness of LAMP. Also, they suggest to add the dye to the inside of the cap of the tube where it remains during the reaction so that one can mix it in later without having to open the tube.

16. The crucial question for LAMP assays is that of specificity. Given that even in hot spots, prevalence is typically only around 1%, we need a specificity above 99% in order to avoid raising more false than true alarms. The authors have tested negative 25 gargle-lavage pools without getting a false positive. The 95% binomial confidence interval for 25/25 is (using Wilson's method) 87-100%. In other words: with only 25 negative samples, it is hard to say that the assay has sufficient specificity.

To really be able to say (with 95% confidence) that the assay has a specificity over, say, 98%, one would need to see no false positive after testing at least ~190 negative samples.

This is, of course, quite an effort. However, without this, the authors' claim that they have better specificity than other recently published approaches cannot be made. This, of course, is not necessary to publish justification, but the text should somehow acknowledge that the available sample size is not sufficient to make too strong claims about specificity (or claims that there are "no" sporadic false positives, as Table 1 claims).

Put is in another way: giving binomial confidence intervals is really important whenever one makes claims about *not* having observed some outcome after having made n tries (as otherwise, papers with smaller will look stronger than those with large n).

Similar issues arise for other claims in Table 1: Does the addition of helicase really improve specificity? Supplementary Figure 3A makes a strong case for this claim, but the extent of this improvement in practice is hard to estimate without very many samples.

This is an excellent suggestion. To address the question of specificity we have now tested a total of 236 SARS-CoV-2 negative samples. If we require both assays (Orf1a gene and N gene) to be

positive, the false positive rate is 0% (95% binomial confidence interval 0 – 1.6%). However, we note that very rarely one of the two assays was positive ($3/235 = 1.3\%$, 95% CI 0.3 – 3.7%). Those three samples were negative for both assays in a second replicate experiment.

17. The same goes for the core result: would a conventional magnetic bead extraction really result in more false positives than the hybridization capture suggested here? (Or have I misunderstood and the advantage rather lies in simplicity? The authors perform only one washing step, and, I believe, usual magnetic bead extraction protocols use several.) One way to clarify this might be to perform RT-qPCR before and after bead purification, as done for Figure 1D, but do this twice: once with beads with specific oligos and once with normal beads.

Conventional magnetic bead extraction binds all nucleic acids (RNA and DNA) irrespective of origin thus separating nucleic acids from unwanted and inhibitory substances. However, this is unlikely to prevent the false positive amplification as we observe it in Supp. Fig. 5. We show that this unspecific amplification is likely due to DNA from the oral microbiome, food or host cells as it can be prevented by prior λ exonuclease treatment that preferentially digests 5'-phosphorylated DNA leaving non-phosphorylated primers and iLAMP product intact (Supp. Fig. 5B). Contrary to our hybridization capture-based RNA extraction which results in removal of non-target nucleic acids, conventional magnetic bead extraction cannot remove DNA that can serve as template for unspecific amplification and thus cause false positives.

Minor issues:

18. - I'm not familiar with SybrGreen's orange-to-yellow colour change: is this caused by the same conformation change as the well known change to become fluorescent? Is it really due to intercalation only and independent of pH? A reference to a paper describing this might be helpful.

We believe it is caused by the known intercalation of SYBR Green I into the minor groove of double-stranded DNA that is normally visualized using fluorescence, but that daylight is sufficient to make it visible by eye when adding 10000x concentrated SYBR Green I solution.

19. - The idea to put the dye into the tube cap sounded familiar, so I searched and found that it was discussed in a few older paper as "closed-tube LAMP" (e.g., <https://doi.org/10.1007/s00705-018-3706-0>). maybe the authors could try to find the original first reference to the idea and cite it?

Thank you for making us aware of this study. We now cite the mentioned publication in the section 'Development of Cap-iLAMP' as follows: 'A similar closed tube approach has been described for detection of yam mosaic virus (Nkere et al. 2018).'

20 and 21. - It's nice that the same two colours are used for negative and positive samples throughout, but I find the chosen bluish green and greenish blue very hard to distinguish. Figure 2F is too busy. I cannot see anything with all these lines. Maybe it could be split into two panels?

We agree that Fig. 2F is hard to read and does not contain vital information so we removed it. Amplification curves of the final iLAMP reaction can now be found in Supp. Fig. 6.

22. - Figure 2C: the bracket should be explained in the figure caption, not only in the main text.

We have added a sentence to the figure legend 'Two positive samples (P-B and P-C) are negative for both assays, but are positive when only 1/20 of input is used thus hinting to inhibition by these samples if not diluted.' (Supp. Fig. 7)

23. - The authors claim that the use of a colour picking app gives absolute and reproducible results independent of subjective viewing preferences. I agree but worry about mobile phone's automatic white balancing functionality. Could the authors please check by measuring the hue of the same sample under different lighting conditions (e.g., fluorescent light from a standard ceiling light versus sunlight outside on a bright day). They may need to caution the user to always use the same lighting conditions.

Thank you for suggesting this additional experiment. We have now tested different light conditions (daylight and fluorescent tube light) and confirm that it is indeed important to use the same lighting

conditions – preferably daylight. We add this observation in the results section: ‘While both smartphones result in comparable hue values, daylight resulted in a clearer separation between negative and positive samples, than fluorescent tube light (Supp. Figure 3 A and B). We thus acquired all images using daylight.’

24. - Figure 1D is hard to read: It would be better to use one axis for the CT value before and the other for the CT value after purification. The viral loads could then be shown as secondary axes.

We believe adding two additional secondary axes might also result in a figure that is hard to read. To improve readability, we added a clarification to the figure legend: ‘Capture efficiency of three samples with different viral loads was estimated relative to automated silica-based RNA extraction based on copy number estimates for RT-qPCR assay targeting the SARS-CoV-2 E gene.’

Overall, I consider this submission to be a quite valuable contribution to the current efforts to find a good and simple test, although the text needs to be worded a bit more carefully at some places, in order to avoid overstating claims.

Thank you for the encouraging words and valuable input.

Reviewers' Comments:

Reviewer #1:

Remarks to the Author:

Most reviewer comments were addressed in the revised version. I am however still of the opinion that the diagnostic value of the method is being oversold given its high false-negative rate, and limitations of the method are not sufficiently discussed. The authors should highlight the limitations (relatively high rates of false results) in the discussion. Considering such phrasings is especially important for a method ultimately meant for medical diagnostics. Classifying SARS-CoV-2 positive samples with RT-PCR $Ct \geq 24$ separately as "non-infectious" based on Bullard 2020 in the analyses seems an ad hoc solution given that Ct 24-29 would not be considered weakly positive samples in RT-PCR methodology. It is also well known that the viral load in the patients fluctuate over time, so that the exact Ct at a given time-point of sampling is not informative on whether a >patient< is infectious or not. In addition the Ct depends on the sampling procedure and sample quality, which can vary even for the very same patient in repeated sampling at the same time point. When classifying samples as positive or negative the authors sometimes considers both probes and sometimes one, while robust criteria are needed for tests in real situations. Altogether I do find the manuscript a good addition to the LAMP field with some potential to help during the covid-19 pandemic, but the discussion needs to be appropriately moderated.

Reviewer #2:

Remarks to the Author:

The authors have revised the text and clarified a few points; they have also added new data concerning specificity. The paper presents the results now in a comprehensive manner.

Of course, it is still a small study, but the sample numbers are now sufficient to back up the claims made.

I'm still a bit disappointed that the data is particularly weak in the one truly novel aspect of the work, namely the use of hybridization capture instead of ordinary RNA amplification. The authors claim that it improves specificity, but cannot say by how much as they don't have numbers for the specificity one would get without.

In the end, we now get a specificity of ~99%. This would be a bit low for a laboratory-based test but is quite decent for a point-of-care test where one could relegate verification to a PCR follow-up in case of a positive.

For the sensitivity, the authors have now set a threshold at $CT \sim 24$ and cite sensitivity for sample below and above this threshold. That is a good way to report these numbers, but 24 is a rather low value: one might hope for nearly perfect sensitivity here, but this only reached in what the authors call the "relaxed" setting.

Overall, these numbers are hence ok, but not great -- but on the other hand certainly honest and not promising too much.

Would such a test help in practice? Where should it be used? I now wonder how the authors numbers compare to other point-of-care tests, such as the currently much discussed rapid antigen tests marketed by various companies.

Response letter

We thank the reviewers for their time and effort. We have addressed their comments in the revised version of the manuscript as detailed below.

Reviewer #1 (Remarks to the Author):

Reviewer report for “Rapid, reliable, and cheap point-of-care bulk testing for SARS-CoV-2 by combining hybridization capture with improved colorimetric LAMP (Cap-iLAMP)” by Bokelmann et al.

SUMMARY:

Most reviewer comments were addressed in the revised version. I am however still of the opinion that the diagnostic value of the method is being oversold given its high false-negative rate, and limitations of the method are not sufficiently discussed. The authors should highlight the limitations (relatively high rates of false results) in the discussion. Considering such phrasings is especially important for a method ultimately meant for medical diagnostics.

We now stress the limitations in the Discussion. We point out that this assay should be used for screening purposes rather than as a diagnostic tool to limit the spread of infections, especially in settings where laboratory equipment are not available. For diagnosis, individual tests, preferably by RT-qPCR, should obviously be performed.

Classifying SARS-CoV-2 positive samples with RT-PCR $Ct \geq 24$ separately as "non-infectious" based on Bullard 2020 in the analyses seems an ad hoc solution given that Ct 24-29 would not be considered weakly positive samples in RT-PCR methodology. It is also well known that the viral load in the patients fluctuate over time, so that the exact Ct at a given time-point of sampling is not informative on whether a patient is infectious or not. In addition the Ct depends on the sampling procedure and sample quality, which can vary even for the very same patient in repeated sampling at the same time point. When classifying samples as positive or negative the authors sometimes considers both probes and sometimes one, while robust criteria are needed for tests in real situations. Altogether I do find the manuscript a good addition to the LAMP field with some potential to help during the covid-19 pandemic, but the discussion needs to be appropriately moderated.

We agree that it is unclear who may be infectious and not. We now introduce the cut-off of Ct 24 as a cut-off between samples that have high viral loads and those that have lower loads. We also cite He et al. 2020 to say that viral loads may fluctuate over time and mention that they may also fluctuate between samples taken on the same occasion (page 5, 3rd paragraph).

Reviewer #2 (Remarks to the Author):

The authors have revised the text and clarified a few points; they have also added new data concerning specificity. The paper presents the results now in a comprehensive manner.

Of course, it is still a small study, but the sample numbers are now sufficient to back up the claims made.

I'm still a bit disappointed that the data is particularly weak in the one truly novel aspect of the work, namely the use of hybridization capture instead of ordinary RNA amplification. The authors claim that it improves specificity, but cannot say by how much as they don't have numbers for the specificity one would get without.

In the end, we now get a specificity of ~99%. This would be a bit low for a laboratory-based test but is quite decent for a point-of-care test where one could relegate verification to a PCR follow-up in case of a positive.

For the sensitivity, the authors have now set a threshold at CT~24 and cite sensitivity for sample below and above this threshold. That is a good way to report these numbers, but 24 is a rather low value: one might hope for nearly perfect sensitivity here, but this only reached in what the authors call the "relaxed" setting.

Overall, these numbers are hence ok, but not great -- but on the other hand certainly honest and not promising too much.

Would such a test help in practice? Where should it be used? I now wonder how the authors numbers compare to other point-of-care tests, such as the currently much discussed rapid antigen tests marketed by various companies.

As mentioned in the Discussion, we propose to use Cap-iLAMP in a variety of settings where rapid on-site screening is crucial.

As suggested by the reviewer, we added a supplementary table comparing the sensitivity and specificity of Cap-iLAMP to antigen tests. We also mention this in the Discussion (page 8, 2nd paragraph from bottom) (specificity is comparable, sensitivity is better for Cap-iLAMP).